# Decoding Algorithm of Motor Imagery Electroencephalogram Signal Based on CLRNet Network Model

**DOI:** 10.3390/s23187694

**Published:** 2023-09-06

**Authors:** Chaozhu Zhang, Hongxing Chu, Mingyuan Ma

**Affiliations:** Department of Electronics Electricity and Control, Qilu University of Technology (Shandong Academy of Sciences), Jinan 250353, China

**Keywords:** motor imagery, EEG, CNN, LSTM, ResNet

## Abstract

EEG decoding based on motor imagery is an important part of brain–computer interface technology and is an important indicator that determines the overall performance of the brain–computer interface. Due to the complexity of motor imagery EEG feature analysis, traditional classification models rely heavily on the signal preprocessing and feature design stages. End-to-end neural networks in deep learning have been applied to the classification task processing of motor imagery EEG and have shown good results. This study uses a combination of a convolutional neural network (CNN) and a long short-term memory (LSTM) network to obtain spatial information and temporal correlation from EEG signals. The use of cross-layer connectivity reduces the network gradient dispersion problem and enhances the overall network model stability. The effectiveness of this network model is demonstrated on the BCI Competition IV dataset 2a by integrating CNN, BiLSTM and ResNet (called CLRNet in this study) to decode motor imagery EEG. The network model combining CNN and BiLSTM achieved 87.0% accuracy in classifying motor imagery patterns in four classes. The network stability is enhanced by adding ResNet for cross-layer connectivity, which further improved the accuracy by 2.0% to achieve 89.0% classification accuracy. The experimental results show that CLRNet has good performance in decoding the motor imagery EEG dataset. This study provides a better solution for motor imagery EEG decoding in brain–computer interface technology research.

## 1. Introduction

Brain–computer interfaces (BCIs) are systems that enable communication between the brain and a device by decoding the electroencephalogram (EEG) signals captured from the brain during mental activities and then generating commands for the device [1,2]. These EEG signals are acquired, processed and classified, typically leading to the conversion into control instructions [3]. These instructions are used to develop systems for neurorehabilitation, recreation and assisting individuals with motor disabilities in achieving independent movement [4]. The flow of the BCI system is depicted in Figure 1. BCI technology has found practical applications in people’s lives, including controlling wheelchairs [5], intelligent prosthetics [6], intentional typing, intentional control of mechanical arms [7,8] and aiding in the recovery from neurological diseases [9], as shown in Figure 2 and Figure 3.

Electroencephalogram (EEG) is a time-varying bioelectrical signal that exhibits characteristics such as temporal variability, low amplitude and high randomness, with a frequency spectrum ranging from 0.5 to 47 Hz. When there is brain activity, electrode recordings can capture the potential changes generated by a large number of neurons. These potential fluctuations over time are commonly referred to as EEG traces. EEG traces reflect the electrical activity of cortical cells in the brain and exhibit complexity and diversity, representing different brain functional states. Motor imagery EEG (MI-EEG) refers to the brain signals generated when an individual imagines body part movements without actually performing them. Compared to traditional EEG signal studies, MI-EEG is better able to reflect people’s intentions and behaviors, and further research on MI-EEG is of significant importance for exploring the neural basis of motor imagery and understanding the underlying neural mechanisms of motor imagery.

The detection of MI-EEG relies on highly sensitive EEG devices and corresponding data analysis techniques. In recent years, there have been significant advancements in the detection and application research of MI-EEG signals, although certain aspects still present challenges. MI-EEG has a low signal-to-noise ratio due to the extremely weak electrical signals obtained by the electrodes when measuring brain neural activity. It is also susceptible to interference from other bodily signals, which necessitates the use of filtering techniques to mitigate the effects of unrelated signals [10]. Furthermore, the non-stationarity of collected MI-EEG signals, influenced by both external environmental factors and internal changes within the subject, introduces unpredictable influences on the acquired MI-EEG, making it highly dynamic [11]. Additionally, since there are individual differences in the collected MI-EEG signals, each participant requires model training from the beginning, resulting in substantial computational costs [12]. Since the features of MI-EEG signals vary over time and can produce large differences among individuals, the selection of reliable and stable feature extraction methods is currently an important research direction [13].

In recent years, deep learning algorithms have developed rapidly and competitively in the field of algorithms, with great success in speech sequence modeling, image classification and video tracking [10]. Deep learning algorithms have powerful end-to-end self-learning capabilities and can automatically extract effective features from complex data. Using deep learning methods to process MI-EEG signals can effectively improve the decoding performance [14,15].

Some researchers proposed separating the channels of CNN to encode multichannel data and then connecting the encoded features back to the recognition network to perform the final MI-EEG recognition task [16]. It was demonstrated that features encoded by CNN are easier to recognize. Similarly, RNN can efficiently learn temporal feature relationships in EEG by virtue of their sequential network structure [17]. The authors of [18] combined LSTM with bi-directional LSTM (BiLSTM) to extract spatial and temporal attributes in the EEGmmidb dataset and improved the classification accuracy by 8.25% compared to the traditional method. Considering the advantages and disadvantages of CNN and RNN, combining CNN and RNN into one network can improve the prediction accuracy of MI-EEG and is more suitable for the feature processing of non-smooth EEG [19]. The authors of [20] combined CNN and LSTM, incorporating both methods into a single network to improve the performance of the network. In [21], an EEG signal classification method based on a multiscale CNN model is proposed and the performance of the network is evaluated on a publicly available EEG dataset. The authors of [22] added attention mechanisms to multi-scale fused convolutional neural networks for visual analysis of EEG signal decoding. The authors of [23] enhanced the time-frequency representation of EEG data using an improved deep convolutional generative adversarial network. The authors of [24] proposed a weighted shared 2D convolutional CNN-LSTM network that shares convolutional kernels of different channel feature maps. The weight-sharing CNN-LSTM reduces the amount of calculation and speeds up the network training, and the highest accuracy rate is 82.3%. The authors of [25] proposed two meaningful image representations built from multichannel EEG signals. Images were built from spectrograms and scalograms. They evaluated two kinds of classifiers: one based on CNN-2D and the other built using CNN-2D combined with LSTM. Their experiments showed that this pipeline allows us to use the same channels and architectures for all subjects, achieving competitive accuracy using different datasets: 71.3 ± 11.9% for BCI IV-2a (four classes).

For the data processing stage of MI-EEG, many effective methods have been proposed by scholars. However, due to the limitations of the data and the characteristics of the EEG signal itself, bottlenecks have been encountered in the current stage of research, and it is difficult to improve the classification results of MI-EEG signals to a large extent. It has been shown that deep learning methods such as CNN and RNN have good results on MI-EEG classification, and the decoding of MI-EEG signals can be better achieved by fusing multiple deep learning algorithms into one network and taking advantage of each network. In terms of classification, research on MI-EEG signals needs to be further improved in the following aspects:(1)Improving algorithmic models: The study of feature extraction algorithms and classification algorithms with better noise immunity is one of the goals to achieve the improved classification accuracy of MI-EEG signals. Better feature analysis can be achieved by using automatic feature extraction methods based on neural networks. Meanwhile, traditional classification models such as Bayesian networks and support vector machines require a large number of training samples to obtain more satisfactory classification results. The algorithmic model to improve the classification accuracy of motion images using limited samples is a key research direction at present.(2)Algorithm combination: Each algorithm has its own advantages and disadvantages. Multiple algorithms can be combined (e.g., multi-feature combination) to leverage the advantages of each algorithm to obtain more comprehensive feature information, thus providing more predictive interpretation and improving the accuracy and generalization of classification. Moreover, since different algorithms have different error responses when faced with the same data, the combination of algorithms can improve the overall model stability and facilitate the extension to new dataset applications.

Based on this, this study proposes a combination of CNN and BiLSTM and ResNet, i.e., CLRNet is used for decoding MI-EEG. CLRNet is a hierarchical end-to-end network structure in which a convolutional neural network (CNN) is used to find the most informative linear subspace in the MI-EEG signals by a layered end-to-end network structure. A special recurrent neural network (RNN) with a bi-directional long short-term memory network (BiLSTM) was then developed as a regression algorithm to capture temporal dynamics. For the long sequence feature information captured by the BiLSTM network, a ResNet cross-layer connection was used to purposefully enhance the data processing capability of the network. This study compares the CLRNet model with various other methods on publicly available datasets and finds that the model constructed in this study performs well on the BCI Contest IV dataset 2a.

## 2. Related Work

### 2.1. Convolutional Neural Network

Neural networks have powerful nonlinear fitting and self-learning capabilities. Compared with traditional methods, neural networks have a powerful network structure and are able to omit the step of manual feature extraction. They have excellent performance in the feature extraction part of network training and in handling classification and regression problems. CNNs reduce the complexity of the feedback network and enhance the generalization ability of the network through unique structural designs, such as locally shared weights and neural-network-based pooling. CNN can extract the rich linear subspace of the original signal and update the weights using a back propagation algorithm, which is one of the main advantages of CNN.

### 2.2. Bidirectional LSTM Network

RNN is used to extract temporal patterns and build a sequential model of the time series. In this model, past data iterations predict future inputs, while future iterations infer past interactions. This type of network is applied in RNN variants of LSTM and gated recurrent neural (GRU) networks. LSTM is an improved version of RNN that increases the unit structure and reduces the weight of unimportant information. It effectively improves the learning efficiency of long sequence data and avoids the gradient disappearance and gradient explosion problems that RNN usually faces. For an input vector $x_{k}=\left[x_{1},x_{2},\cdots ,x_{k}\right]$ of length k, the LSTM computes the next sequence of hidden vectors $h_{t}$ by iterating over time.
(1)$$f_{t}=\sigma (W_{xf}x_{t}+W_{hf}h_{t-1}+b_{f})$$
(2)$$i_{t}=\sigma (W_{xi}x_{t}+W_{hi}h_{t-1}+b_{i})$$
(3)$$c_{t}=f_{t}*c_{t-1}+i_{t}*tanh(W_{xc}x_{t}+W_{hc}h_{t-1}+b_{c})$$
(4)$$o_{t}=\sigma (W_{xo}x_{t}+W_{ho}h_{t-1}+b_{o})$$
(5)$$h_{t}=o_{t}*tanh(c_{t})$$

In the above equation, $W_{x}$ and $W_{h}$ denote the trainable weights of the input vector and the cyclic connection, respectively, and b represents the deviation terms. The oblivion gate, input gate, output gate and cell state are denoted by f, i, o and c respectively.

### 2.3. Residual Network

Ideally, the fitting power of a neural network increases with the number of network layers. However, in practice, the cumulative multiplicative effect of the backpropagation algorithm leads to problems in error gradient transfer between shallow and deep networks, resulting in reduced gradients that may disappear or explode. As a result, not enough feature information is extracted to obtain better performance. This is the phenomenon of underfitting as well as overfitting in the network. To solve these problems, ResNet was proposed as one of the classical network structures in the field of deep learning. The innovation of ResNet is the introduction of residual blocks, which combine the outputs of different network layers by spanning connections to form a residual mapping. This design ensures that the gradients can be passed efficiently even if the network is deepened, making the network easier to train and optimize. At the same time, the spanning connection also maintains the flow of gradient information from the previous layer, effectively avoiding the problem of gradient disappearance and dispersion. Its basic structure is shown in Figure 4.

In Figure 4, x is the input of the pre-network layer, relu is the activation function and f(x) is the output of the single-layer network, which form a simple residual network. H(x) is the output of the two network layers and a constant mapping of the initial input through the residuals. A simple residual network is formed.

## 3. Deep-Learning-Based Neural Network Structure Building

The convolutional layer in CNN performs spatial filtering operations on the input MI-EEG signals, optimizing the convolutional effect by adjusting various parameters of the CNN. A pooling layer is added after each convolutional layer to downsample the data, followed by another convolution to obtain deeper features. To mitigate overfitting, dropout regularization is employed. Considering the characteristics of MI-EEG, constructing a CNN structure suitable for processing its specific data format can facilitate feature extraction and dimensionality reduction of EEG. The CNN network structure in this study consists of a total of six layers, including four convolutional layers and two pooling layers. The specific structure is illustrated in Figure 5.

The details of each network layer structure in Figure 5 are as follows:

Input layer: The MI-EEG data format of 22 × 288 × 240 is used as the CNN network input.

Convolutional layers: The first two convolutional layer filters are set to 128, the convolutional kernel size is set to 3 × 3 and the number of filters in the last two convolutional layers is set to 256. The size of the convolutional kernel is set to 5 × 5 and the step size is 1. The features of the input data are obtained by sliding the convolutional kernel window according to the step size and are filled with the same after each convolution.

Pooling layer: The role of the pooling layer is to downsample the input data and reduce the complexity of the overall model computation. Similar to a convolutional layer, a pooling kernel is used to pool the convolutional output of the previous layer, and a maximum pooling approach is taken to compare all data in the pooled region. The output of the pooling area is the largest of these values, the pooling kernel is set to 3 × 3, the step size is 1 and the same padding is used to fill the same after the pooling operation.

Dropout layer: The role of this layer is mainly to prevent overfitting when training the network, and a dropout rate of 0.5 is used to improve the overall generalization ability of the network.

Fully connected layer: The fully connected layer is the last layer of the CNN, and the role of this layer is to integrate the features obtained from the previous network through multiple convolution and pooling. The extracted features are mapped to the sample data to expand the differentiation of the data, and the purpose of classification is achieved using the softmax function.

In CLRNet, the structure and parameters of CNN are introduced. The network parameters and structure of BiLSTM as well as ResNet are shown in Figure 6 and Table 1.

In Figure 6, the network structure of the five-layer BiLSTM is shown, which includes B1, B2, B3, B4 and B5. The same parameters are set for this five-layer network structure, the number of hidden neurons is set to 100 and tanh is used as the activation function. The output for BiLSTM is four classification results, as shown in Table 1. ResNet is added to BiLSTM, and the residual structure of ResNet is applied to the BiLSTM network for cross-layer connectivity. For BiLSTM, the output of the single-layer network is processed through the residuals, i.e., RES1=(B1+B3) as well as RES2=(RES1+B5). The final residual-processed output is verified by softmax for the four classifications. Adopting the cross-layer connection of ResNet can reduce the problem of increasing training error caused by the deepening of the number of layers of the network, and further improve the generalization ability of the network to avoid the occurrence of overfitting. Therefore, it has better performance for processing complex motor imagery patterns. The parameters in the model are optimized after several experiments and adjustments. The parameters presented are validated in later experiments.

CNN performs well in the space domain feature extraction ability, and LSTM can effectively extract information in the time domain. The network model combining CNN and BiLSTM can decode MI-EEG signals more effectively. When the number of layers of the network increases, gradient disappearance and overfitting problems occur. Therefore, there is still room for improvement in the structural optimization of deep networks. In this case, adding ResNet is a very effective way to improve the performance of the network. Enhancing network generalization by adding direct mapping between the outputs of BiLSTM feature vectors using ResNet’s cross-layer connectivity in BiLSTM further enhances the performance of the network in MI-EEG signal decoding tasks.

In the decoding of MI-EEG, preprocessing is necessary to extract the required electroencephalogram data. In this study, we designed a network model that takes preprocessed EEG data as the input for feature analysis and validates four types of motor imagery states, as shown in Figure 7. The overall model architecture combines the spatial convolutional layer of CNN, the time-frequency analysis layer of BiLSTM and the cross-layer connections of ResNet.

When analyzing MI-EEG, it is necessary to continuously explore network structures suitable for its decoding. As both CNN and LSTM have shown promising results in MI-EEG processing, this paper makes improvements on their base network architectures to further enhance the decoding performance of MI-EEG. The proposed CLRNet network structure in this study consists of multiple different network layers. The first six layers are CNN layers used to capture the spatial information of MI-EEG signals, while the following five layers are BiLSTM layers used to capture the temporal information of MI-EEG signals. In the BiLSTM layers, ResNet’s cross-layer connections are incorporated for residual processing, enhancing the network’s data processing capability and stability, and improving its performance. The overall network model combines spatial convolutional transformations, time-frequency analysis and feature selection.

## 4. Experimental Results and Analysis

### 4.1. Experimental Data

When comparing different network models, the training and testing datasets used for the network models under study are often different. Differences in datasets may lead to differences in the accuracy of network models, resulting in some network models outperforming others on some datasets. To address this issue, this study uses the motor imagery EEG dataset commonly used by researchers as the BCI Competition IV dataset 2a. The dataset is obtained from experiments performed on nine experimenters with a sampling rate of 250 Hz, which has been processed using a band-pass filter of 0.5–100 Hz. The dataset contains 25 channels (22 EEG and 3 EOG) and is a four-class MI (left/right hand, foot and tongue) EEG dataset. The paradigm of the dataset is shown in Figure 8.

The first 2 s of the paradigm in the figure is the preparation time. A cue is given at the 2 s mark. From 3 s onwards, subjects performed imaginary movements until the 6 s mark.

### 4.2. Data Processing

In the whole MI-EEG signal acquisition experiment, the experimental data are recorded throughout. According to the introduction of the experimental paradigm, the duration of motor imagery is from 3 s to 6 s. Here, 3 s of the motor imagery segments is extracted via segmentation as the data for this experiment. It is filtered from 8–30 Hz and the effects of the oculomotor signals from the ‘EOG-left’, ‘EOG-central’ and ‘EOG-right’ channels are removed at the same time.

After the operation of filtering and segment extraction of the MI-EEG data, the MI-EEG signals format of a single subject is 22 × 288 × 750. However, inputting the data into the CLRNet network model to verify the performance of the model revealed that the classification accuracy obtained is low. After exploration and experimental validation, it is determined that the initial MI-EEG signal features are relatively weak. It is difficult to extract effective features directly through the CLRNet network model, so the results obtained after feature extraction and classification are weak. To address this problem, a step of simple wavelet packet decomposition is added to the preprocessing stage and good results are achieved. The specific operation is a five-layer wavelet packet decomposition for a sampling rate of 250 Hz. The band range of each node in layer 5 is 0–7.8125 Hz, and the band ranges of nodes 2 and 3 are 7.8125–15.625 Hz and 15.625–23.4375 Hz, respectively. Node 2 and node 3 associated with the Mu (8–14 Hz) and Beta (16–24 Hz) rhythms of motor imagery are selected for signal reconstruction. The format of the reconstructed signal data is 22 × 288 × 240, where 22 represents the number of channels, 288 represents the number of trials and 240 represents the sampling points after data processing. This data format is used for experimental validation in the following experimental procedures.

The CLRNet network model proposed in this study is tested on the BCI Competition IV dataset 2a. The input EEG data format is 22 × 288 × 240. The size of the batch training is set to 8. The number of iterations is set to 200. The Adam optimizer is selected to update the learning weights of the network model, and four categories of motor imagery patterns (left/right hand, foot and tongue) are classified. The dataset used in this study is the BCI Competition IV dataset 2a, which contains data on a total of nine subjects. The nine sets of pre-processed data are validated by the model separately, and the accuracy rate is used as the evaluation index of the classification results. The obtained classification results are shown in Table 2.

The final classification accuracy for each subject is given in Table 2. Individually by subject, the MI-EEG dataset collected through subject 6 is the best classified with an accuracy of 93%. This is the highest classification accuracy for this experiment. Meanwhile, the MI-EEG dataset collected through subject 4 had the worst classification result, which is 83.4%. It is normal that there is variation between subjects due to differences in individual body composition and status. The network model constructed in this study achieved good classification results for all nine subjects’ EEG data from the BCI Competition IV dataset 2a, with an average accuracy of 89%. A high classification accuracy is achieved overall. The results of this experiment show that the model is suitable for processing the EEG signals of most subjects and is effective and practical for processing MI-EEG signals.

In the decoding of MI-EEG signals, feature extraction plays a very critical role, and channel selection is a special feature selection method. Channel selection is closely related to the physiological context in which the MI-EEG signal is located. By removing channels that contain noise and redundant information, the amount of data that needs to be processed can be reduced and the data processing capacity of the BCI system can be improved. Also, adding more valuable channels can improve the signal-to-noise ratio of the system and increase the decoding performance of the BCI system. Therefore, the role of channel selection is very important in the MI-EEG signal decoding task. It can improve the signal quality and interpretability by reducing noise and redundant information and optimize the training and testing accuracy of feature extraction and classifier. In turn, the overall performance of the BCI system is improved. C3 and C4 are considered to be the most relevant channels to MI-EEG [26]. In order to further verify the effect of multiple channels and these two channels on the recognition rate of MI-EEG, these two channels are selected for feature extraction and classification while retaining the C3 and C4 channels. In addition to this, the recognition results are compared with those performed by the three-channel as well as the multi-channel methods using C3, C4 and Cz. The results are shown in Figure 9.

Figure 9 demonstrates the variability between the different channels for the classification accuracy of the subjects. For the MI-EEG data collected from nine subjects, the C3 and C4 channels are most closely associated with MI-EEG. However, adding other channels containing MI-EEG information can enhance the classification accuracy of the model.

In addition, the model proposed in this study contains several commonly used networks. In order to verify that the effect of the combination can obtain the advantages of each network, the CLRNet model proposed in this study is subjected to controlled variable experiments using CNN, BiLSTM and CNN-BiLSTM networks for the BCI Contest IV dataset 2a, respectively. The performance of each network is verified, and then ResNet is added to compare and evaluate the performance of each network model, as shown in Table 3.

Table 3 shows the comparison results obtained from the controlled variable experiments. The experimental results show that the deep learning methods all have good results for the data processing ability of MI-EEG signals. For non-stationary MI-EEG signals, CNN and BiLSTM have good performance for feature extraction and classification of MI-EEG signals. It proves that the spatial and temporal feature information is an important basis for decoding MI-EEG signals.

For non-stationary MI-EEG signals, CNN and BiLSTM have good performance for feature extraction and classification of MI-EEG signals. Their average classification accuracies reached 83.0% and 76.2%, respectively. It proves that spatial and temporal feature information is an important basis for decoding MI-EEG signals. CNN-BiLSTM combines the advantages of both networks and provides excellent performance in processing data containing spatio-temporal information. The average accuracy of the classification is 87.0%, and the overall performance is stronger than that of the individual networks. The CLRNet model proposed in this study combines the features of three networks, CNN, LSTM and ResNet. CNN can effectively extract the spatial features of the signals; BiLSTM can handle sequence information very well. ResNet networks can alleviate the gradient disappearance problem in deep neural networks and reduce the complexity of network training. The experimental results show that the combined approach of the CLRNet model can give full play to the advantages of each network and realize the improvement of classification accuracy of BCI race IV dataset 2a. It is further demonstrated that the CLRNet model proposed in this study is an effective method for MI-EEG signal decoding, and it also shows that the multi-network fusion model is an important research direction for MI-EEG signal processing.

To demonstrate the excellent decoding performance of the CLRNet network model for MI-EEG in more depth, this paper is compared with MI-EEG classification methods proposed by other researchers. All comparison trials are conducted on the dataset of each subject in the BCI Competition IV dataset 2a. The same classification accuracy is used as an evaluation metric. The comparison results are shown in Table 4.

The four MI-EEG classification methods included in this paper are compared separately in Table 4. Temporal convolutional networks (TCNet) and attention-based temporal convolutional networks (ATCNet) are characterized by the ability to achieve better classification results using fewer training parameters. It is suitable for low computational complexity and low memory consumption but has the disadvantage that the extracted features are not comprehensive enough. Wavelet Decomposition-Common Spatial Pattern Algorithm-Artificial Neural Network (WPD_CSP_ANN) is characterized by good performance in combining neural networks for MI-EEG recognition through feature extraction in the time and space domains, but the network design in the feature extraction phase is more complex. The comparison shows that the classification accuracy of the model in this paper reaches 89%, which is significantly higher than the remaining three groups. In this paper, CNN, BiLSTM and ResNet are combined to obtain higher feature recognition capability through feature extraction in multiple dimensions. Compared with other methods for non-smooth MI-EEG signal processing, it has better decoding performance and improves the classification accuracy of MI-EEG signals.

## 5. Discussion

In this research work, a network consisting of CNN, BiLSTM and ResNet is proposed to decode MI-EEG. The main elements of this study are as follows: A hierarchical network is proposed, combining CNN and BiLSTM with ResNet. CNN is used to obtain the null domain features from the original MI-EEG, and BiLSTM is used to obtain the time domain information of the MI-EEG time series. To avoid network degradation, a residual network is used to perform residual processing on the output of BiLSTM to enhance the data processing capability of the model. The CLRNet model presented in this study is evaluated using a publicly available motor imagery dataset to demonstrate the validity and utility of the model. The experimental results demonstrate the ability of CLRNet to effectively reduce the long-term non-smooth effects of EEG sequences and improve the robustness and accuracy of EEG-based motor imagery classification models. The model may be used to develop MI-BCI applications in the future.

For neural networks, the settings of network parameters are not fixed, and different parameters may bring results with large disparities. In order to achieve better MI-EEG decoding, there may be room for improvement for the CLRNet network model architecture parameters proposed in this study. This requires a lot of time for tuning and comparison, and this is the area that needs to be focused on and studied in the future.

## Figures and Tables

**Figure 1 sensors-23-07694-f001:**
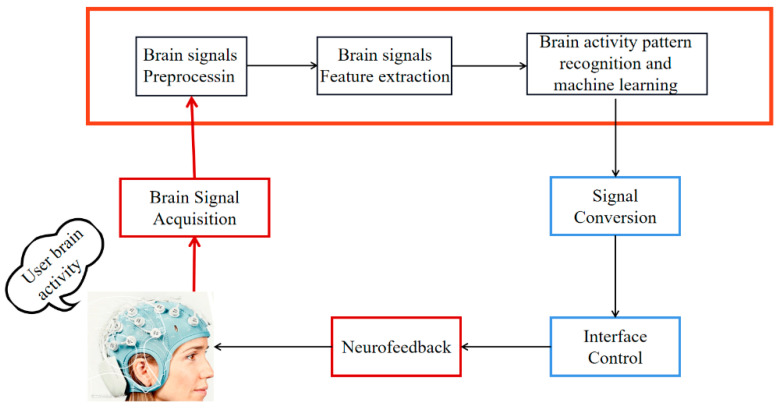
BCI system.

**Figure 2 sensors-23-07694-f002:**
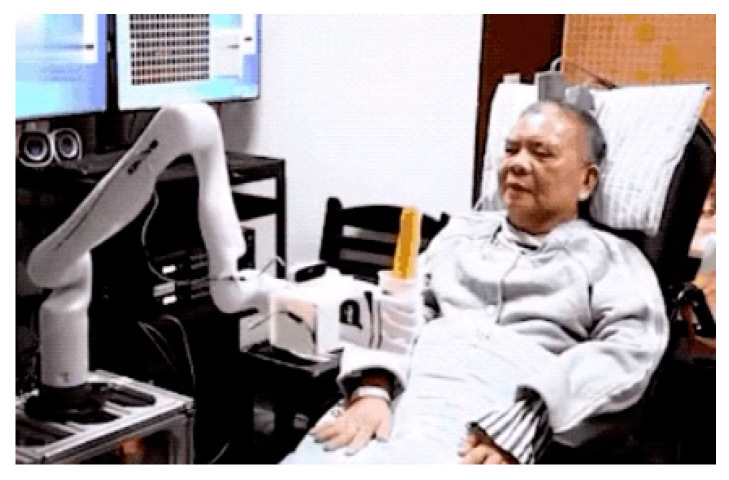
Patient item grasping through the MI-BCI system.

**Figure 3 sensors-23-07694-f003:**
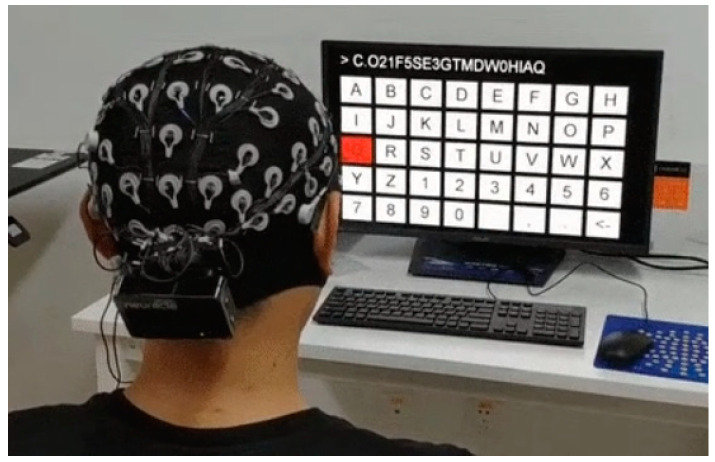
Brain-controlled character input based on the BCI system.

**Figure 4 sensors-23-07694-f004:**
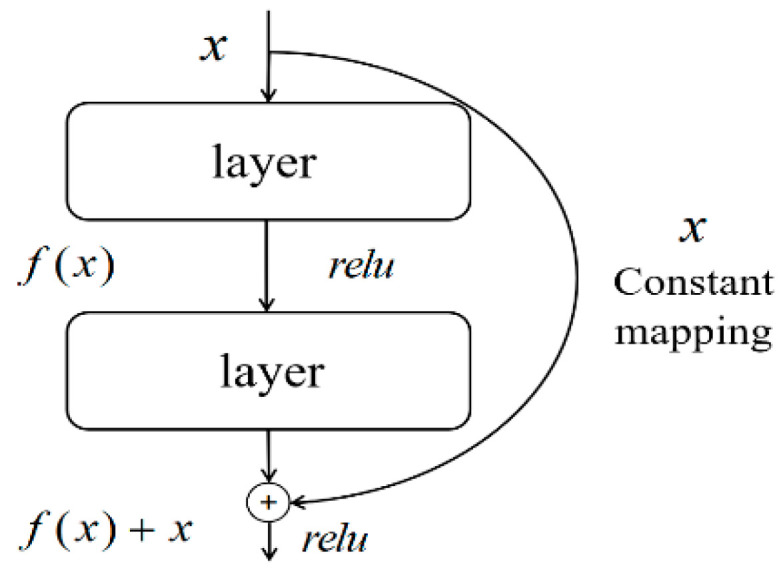
Residual learning unit.

**Figure 5 sensors-23-07694-f005:**
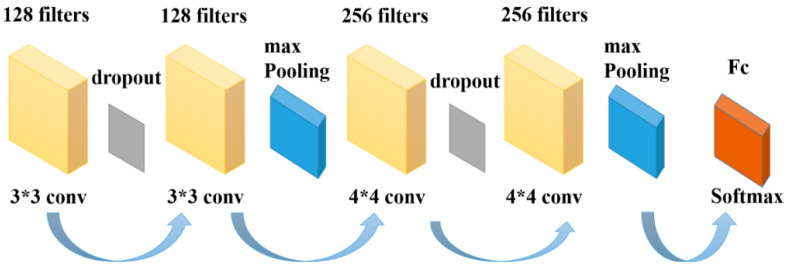
CNN structural parameters.

**Figure 6 sensors-23-07694-f006:**
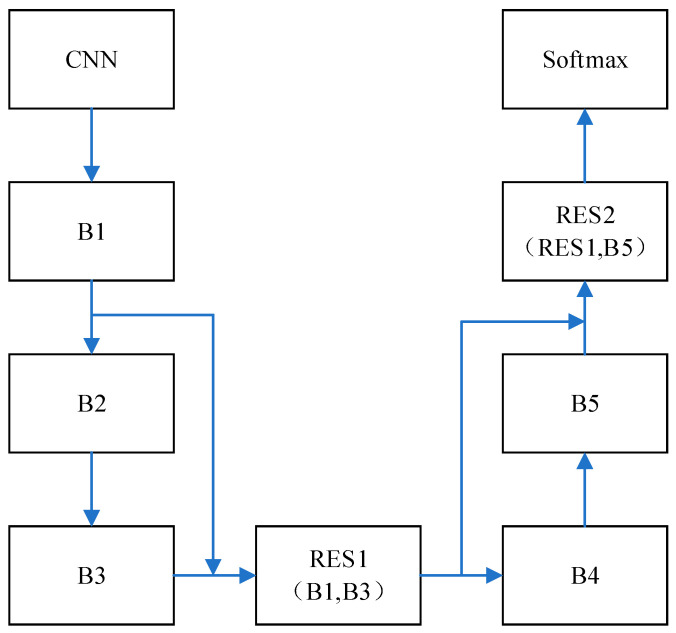
BiLSTM network structure diagram.

**Figure 7 sensors-23-07694-f007:**
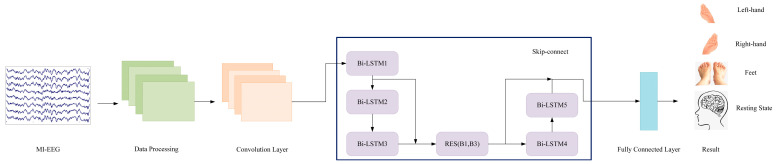
CLRNet network model diagram.

**Figure 8 sensors-23-07694-f008:**
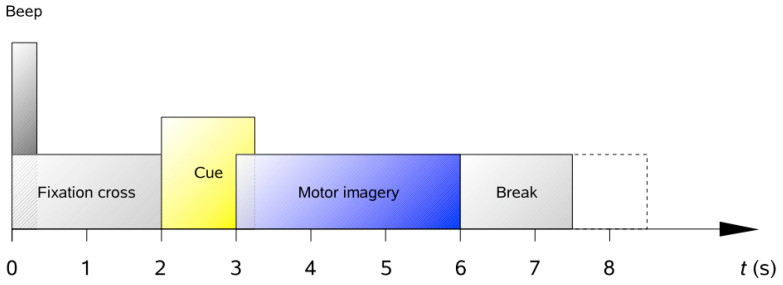
Paradigm’s timing scheme.

**Figure 9 sensors-23-07694-f009:**
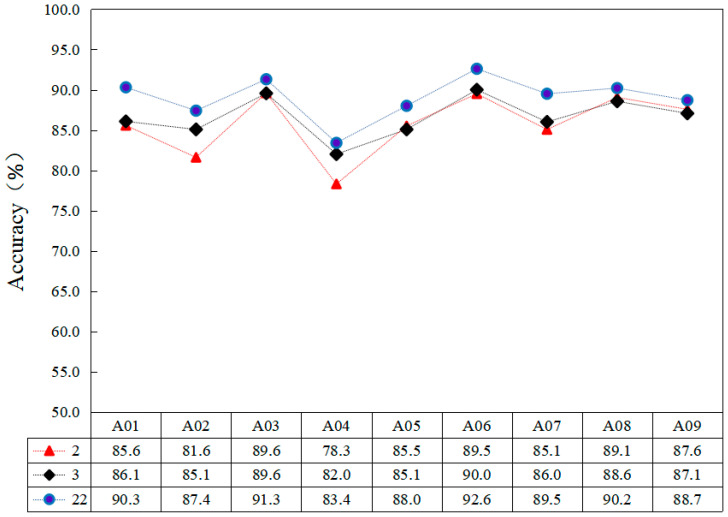
Effect of different number of channels on subjects’ accuracy.

**Table 1 sensors-23-07694-t001:** BiLSTM network parameters.

BiLSTM Network Parameters	Value
Number of networks	5
Cell	100
Activation function	tanh
Return_sequences	Ture
Dense	4

**Table 2 sensors-23-07694-t002:** CLRNet network model classification results.

Subjects	Accuracy (%)
1	90.3
2	87.4
3	91.3
4	83.4
5	88.0
6	92.6
7	89.5
8	90.2
9	88.7

**Table 3 sensors-23-07694-t003:** Controlled variable experiments.

Methods	Accuracy of Different Subjects (%)									Average Accuracy Rate (%)
	A01	A02	A03	A04	A05	A06	A07	A08	A09	
CNN	81.2	79.8	84.6	80.2	82.0	85.1	85.0	84.7	84.5	83.0
BiLSTM	77.0	72.3	78.1	72.1	76.7	78.8	77.0	76.9	77.3	76.2
CNN-BiLSTM	86.8	83.5	88.9	82.8	88.0	89.5	86.8	87.0	87.1	87.0
CLRNet	90.3	87.4	91.3	83.4	88.0	92.6	89.5	90.2	88.7	89.0

**Table 4 sensors-23-07694-t004:** Comparison of MI-EEG classification methods.

Subjects	TCNet (%)	WPD_CSP_ANN (%)	ATCNet (%)	CLRNet (%)
A01	86.1	70.5	88.5	90.3
A02	66.0	60.0	70.5	87.4
A03	93.4	81.8	97.6	91.3
A04	72.6	65.6	81.0	83.4
A05	79.9	75.8	83.0	88.0
A06	66.7	72.0	73.6	92.6
A07	90.3	71.3	93.1	89.5
A08	85.8	71.5	90.3	90.2
A09	85.4	81.0	91.0	88.7
Average	80.7	72.2	85.4	89.0

## Data Availability

The data presented in this study are available in https://www.bbci.de/competition/iv/ (accessed on 19 June 2023).

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
