# Peer review of "Decoding Algorithm of Motor Imagery Electroencephalogram Signal Based on CLRNet Network Model"

_sensors, 2023, doi:10.3390/s23187694_

Round 1
Reviewer 1 Report
The overall contribution and novelty of the work are marginal, it is unclear what is innovation of the work as most of the materials are based on exiting methods, rendering it unsuitable for the journal to further consider its possible publication. Thus, this reviewer doesn't find this manuscript suitable for publication due to its limited methodological novelty and contribution as it does not enhance the existing body of knowledge in the given subject area. Neither the proposed method nor the application is novel. Therefore, I strongly recommend reject decision for this paper.
Good
Reviewer 2 Report
This work explores EEG decoding in motor imagery for brain-computer interface technology. Traditional methods rely on preprocessing and feature design, while deep learning models have shown promise. The study proposes a CNN-LSTM network, CLRNet, integrating ResNet for cross-layer connectivity. CLRNet achieves 89.0% accuracy in classifying motor imagery patterns, demonstrating its effectiveness in decoding EEG datasets and providing an improved solution for brain-computer interface research. Followings are my concerns:
1. There are some misprints. For example, "Figure1" (should be Figure 1), "Figure2" ......
2. In order to underscore the significance of the research topic, it would be beneficial to conduct a thorough review and incorporation of pertinent related works. E,g, EEG-based emotion recognition using hybrid CNN and LSTM classification, Frontiers in Computational Neuroscience.
3. The authors need to provide a more effective demonstration of the contributions and motivations underlying the proposed scheme.
4. There is room for improvement in the language used throughout the manuscript.
5. Additional explanations for Figure 7 are anticipated to provide a clearer understanding of its content and implications.
Minor editing of English language required.
Reviewer 3 Report
The paper titled "Decoding Algorithm of Motor Imagery EEG Signal Based on CLRNet Network Model" presents a contribution to the field of brain-computer interface technology. The study focuses on the important task of decoding EEG signals based on motor imagery, which plays a crucial role in brain-computer interface performance.
The study is well-structured, and the English is generally fine, with only a few minor errors. However, I have some suggestions for improvement.
"contains a large amount of physiological, psychological and pathological information."
Regarding the statement that the EEG contains a large amount of physiological, psychological, and pathological information, it is necessary to clarify that the EEG can provide information related to physiological, psychological, and pathological aspects of a subject, although these terms are broad and vague.
"Motor Imagery(MI), a classical brain-computer interface paradigm, indicates that imagined bodily movements generate (8-14 Hz) rhythmic and (16-24 Hz) rhythmic signals in the motor sensory centers of the cerebral cortex, i.e, event-related synchronization (ERS)/de-synchronization (ERD) phenomenon[11]."
The authors introduce Motor Imagery (MI) as a classical brain-computer interface paradigm, indicating that imagined bodily movements generate rhythmic signals in the motor sensory centers of the cerebral cortex, known as the event-related synchronization (ERS)/desynchronization (ERD) phenomenon. It is important to note that ERD/ERS are not solely rhythmic activity but rather relative changes in the power or amplitude of specific frequency bands, such as mu and beta, in the EEG during motor imagery.
"MI-EEG signals are characterized by nonsmoothness"
Regarding the statement that MI-EEG signals are characterized by nonsmoothness, the term "nonsmoothness" is ambiguous. It would be beneficial for the authors to rephrase or provide an explanation/definition to clarify this characterization.
"For MI-EEG feature extraction, the Common Spatial Pattern (CSP) and the variant Filter Bank Common Spatial Pattern (FBCSP) based on it are commonly chosen in the traditional methods[13,14]. Since the features of MI-EEG signals vary over time and can produce large differences among individuals. "
I appreciate the authors for addressing the issue of changes in MI-EEG over time, which is often overlooked. I would suggest that the authors expand their literature review to encompass the topic of nonstationarity or temporal changes, as these factors can significantly impact the feature extraction process and result in non-stationary feature covariance shifts. This concern has been discussed in a relevant study (10.1016/j.cmpb.2020.105808) and should be taken into consideration. The presence of non-stationary data violates the assumptions of most machine learning approaches currently available, which assume that the features are stationary. This is particularly important in BCI MI applications, where there is typically an initial "calibration session" for classifier training, followed by an "online" session that may be 20-30 minutes apart. Any changes in the means and standard deviations of the features between these sessions can significantly affect performance. It is worth noting that these shifts are more pronounced when using wet electrodes, due to the chemical reaction between saline and metal electrodes, as well as the drying of the gel over time.
General comments on the discussion:
While the presented results are promising, it is essential for the authors to discuss the potential limitation of the system. Specifically, the extraction of features considering the entire 3 seconds of motor imagery may not align with the time-critical nature of BCI systems that involve human interaction. The delay of 3 seconds between the onset of motor imagery and obtaining a response is typically unacceptable for users, as it affects their perception of being in control. Therefore, it would be valuable for the authors to address this issue and propose ideas to improve the time-critical perspective.
Furthermore, it would be beneficial to compare their system with more traditional approaches mentioned in the introduction, especially in terms of dealing with noisy and nonstationary EEG data.
Overall, the work is well-executed, with the mentioned improvements, that could contribute to the advancement of brain-computer interface technology.
Minor editing of the English language is required, mostly in the introduction.
Round 2
Reviewer 1 Report
The authors have done a good job.
Reviewer 3 Report
I would like to thank the authors for their responses and the new version submitted
Responses to authors:
Firstly, what we want to show about the claim that the MI-EEG signal is nonstationary is that, MI-EEG signals are characterized by nonsmoothness due to various reasons such as the dynamic nature of motor imagery, subject-specific differences, noise and artifacts, spatial distribution of brain activity, and modulation of oscillatory activity. These factors contribute to the irregular patterns observed in the EEG signals during motor imagery tasks.
Indeed, I concur with your viewpoint. However, since the current discussion lacks depth, I believe it would be beneficial to explore the topic further. I propose addressing this as a challenge or issue in motor imagery and EEG in general, backed by relevant literature to support our assertions (see previous comments)
Secondly, regarding your question about the 3-second delay. Because the motion imagining time provided by the public dataset is three seconds so we did the feature extraction according to three seconds. In subsequent work we will try to shorten this time to achieve compliance with the time criticality of BCI systems involving humancomputer interaction.
Kindly address this issue in the appropriate section. It is a crucial matter that is frequently overlooked, with performance often solely assessed based on classification performance. The reason behind requesting a discussion on these aspects is to raise awareness that pure accuracy is inadequate to determine the superiority of a particular BCI or any real-time system. I invite the authors to deliberate on these issues and consider them as potential objectives for their future studies.
Minor issue
Line 267
To address this issue, this chapter uses the motor imagery EEG dataset commonly used by researchers as the BCI Competition IV dataset 2a[].
There appears to be a typographical error, as the reference is missing.
